# Picocyanobacteria and deep-ocean fluorescent dissolved organic matter share similar optical properties

Zhao Zhao[1,2], Michael Gonsior[3], Jenna Luek[3], Stephen Timko[4], Hope Ianiri[5], Norbert Hertkorn[6], Philippe Schmitt-Kopplin[6,7], Xiaoting Fang[8], Qinglu Zeng[9], Nianzhi Jiao[1,2] & Feng Chen[2,10]

Marine chromophoric dissolved organic matter (CDOM) and its related fluorescent components (FDOM), which are widely distributed but highly photobleached in the surface ocean, are critical in regulating light attenuation in the ocean. However, the origins of marine FDOM are still under investigation. Here we show that cultured picocyanobacteria, *Synechococcus* and *Prochlorococcus*, release FDOM that closely match the typical fluorescent signals found in oceanic environments. Picocyanobacterial FDOM also shows comparable apparent fluorescent quantum yields and undergoes similar photo-degradation behaviour when compared with deep-ocean FDOM, further strengthening the similarity between them. Ultrahigh-resolution mass spectrometry (MS) and nuclear magnetic resonance spectroscopy reveal abundant nitrogen-containing compounds in *Synechococcus* DOM, which may originate from degradation products of the fluorescent phycobilin pigments. Given the importance of picocyanobacteria in the global carbon cycle, our results indicate that picocyanobacteria are likely to be important sources of marine autochthonous FDOM, which may accumulate in the deep ocean.

[1] State Key Laboratory for Marine Environmental Science, Institution of Marine Microbes and Ecosphere, College of Ocean and Earth Science, Xiamen University, Xiang'an Campus, Xiang'an South Road, Xiamen 361102, China. [2] Institute of Marine and Science Technology, Shandong University, Joint Lab of Microbial Oceanography at QNLMST, Wenhai Road, Qingdao 266237, China. [3] Chesapeake Biological Laboratory, University of Maryland Center for Environmental Science, 146 Williams Street, Solomons, Maryland 20688, USA. [4] Department of Civil and Environmental Engineering, University of California Irvine, E4130 Engineering Gateway Building, Irvine, California 92697, USA. [5] Department of Chemistry and Chemical Biology, Northeastern University, 360 Huntington Avenue, Boston, Massachusetts 02115, USA. [6] Helmholtz Zentrum Muenchen, Deutsches Forschungszentrum für Gesundheit und Umwelt, Research Unit Analytical BioGeoChemistry, Ingolstaedter Landstrasse 1, 85764 Neuherberg, Germany. [7] Analytical Food Chemistry, Technische Universität München, Alte Akademie 10, 85354 Freising, Germany. [8] Environmental Science Programs, School of Science, The Hong Kong University of Science and Technology, Clear Water Bay, Kowloon, Hong Kong, China. [9] Division of Life Science, The Hong Kong University of Science and Technology, Clear Water Bay, Kowloon, Hong Kong, China. [10] Institute of Marine and Environmental Technology, University of Maryland Center for Environmental Science, 701 East Pratt Street, Baltimore, Maryland 21202, USA. Correspondence and requests for materials should be addressed to M.G. (email: gonsior@umces.edu).

Marine picocyanobacteria, mainly *Synechococcus* and *Prochlorococcus*, are abundant and widely distributed unicellular prokaryotic phytoplankton in the global marine environment[1,2]. As major primary producers in the World's Oceans[3], picocyanobacteria also contribute to the marine dissolved organic matter (DOM) pool in the surface ocean[4,5] and to particles sinking to the deep ocean. Marine open ocean sediment trap records showed strong cyanobacterial pigment signals in deep-ocean samples well below the mixed layer[6]. Vertical mixing[7] and packing of cells into grazers' faecal pellets are important pathways to transport picocyanobacterial cells directly from the photic zone to the deep sea[8,9]. *Synechococcus* cells were estimated to contribute 0.5–6.6% of the total marine particulate organic matter flux through protozoa to mesozooplankton faecal pellets[9]. Lysis and degradation of picocyanobacterial cells during the vertical transport supply DOM to the abyssal ocean, but picocyanobacteria-derived DOM and its light-absorbing chromophoric DOM (CDOM) component have not been previously characterized in detail and have not been directly compared to CDOM present in the deep ocean.

A well-established typical feature of deep-sea DOM is its increased CDOM fraction relative to surface DOM, including a subset of CDOM that has fluorescent properties (fluorescent DOM; FDOM) with large Stokes shifts and similarities to terrestrially derived FDOM. The increased abundance of specific FDOM components with depth is also a well-characterized feature of marine DOM[10–16] that has been often characterized using excitation emission matrix (EEM) fluorescence coupled with Parallel Factor Analysis (PARAFAC)[17–21]. It is also well established that photo-degradation in the surface ocean depletes CDOM, including FDOM[10,22,23]. Despite its photobleaching in the surface ocean, CDOM and FDOM still play a major role in controlling light penetration and protection of marine biota from harmful ultraviolet irradiation[24]. While the vital part of CDOM in the marine carbon cycle has been readily acknowledged, diverse sources have been discussed in recent years[13,15,21,25,26]. Terrestrial-derived material, *in situ* production from marine biota and sediment leaching are all considered to represent important sources of marine CDOM[27]. However, to the best of our knowledge, a very close match of ultraviolet–visible (UV–Vis) absorbance, EEM fluorescence, its photochemical degradation behaviour, as well as apparent fluorescent quantum yield (QY) comparisons between marine sources and deep-sea CDOM have not yet been established.

While terrestrially derived CDOM is largely removed in the coastal transition zone[28,29], low concentrations of lignin derivatives, as tracers for terrestrial CDOM, can still be found in the coastal and open ocean[30,31]. However, average cycling times of this terrestrial CDOM were estimated to be only ∼90 years[32], which is far too short to explain the widespread occurrence of deep-sea CDOM in the World's Oceans, but the very good spectral match between terrestrial-derived and deep-sea CDOM puzzled scientists in the last decades.

The *in situ* production of marine CDOM has been discussed for some time, and linking marine biota with specific optical properties (for example, EEM fluorescence) has recently been explored in more detail, but did not find specific marine sources of this material. A strong correlation between FDOM fluorescence with apparent oxygen utilization was demonstrated for the Pacific Ocean suggesting that this relationship is indicative of *in situ* production of bio-refractory FDOM at depth[12]. However, other studies found that this correlation does not hold in the Atlantic Ocean[14], further fuelling the discussion about sources.

The correlations between oceanic CDOM distribution and chlorophyll concentrations[33] were also considered to be significant indicators of marine autochthonous CDOM production; however, no such relationship has been demonstrated between marine subsurface chlorophyll maxima and CDOM. Previous studies investigated the release of CDOM from different phytoplankton and showed a rather small release of CDOM by diatoms[34] that may explain up to 20% of the CDOM signals seen in some diatom dominated areas. It also was shown in another study of 11 algal cultures that phytoplankton did not contribute to the CDOM[25] pool and that CDOM might be generated by heterotrophic bacteria. Some evidence of low production of marine FDOM was given from dark incubation experiments[35–37], but cell abundances and bacterial community structures were not evaluated. While contradictory results have been described for eukaryotic algal species, the production of CDOM has been shown in colonial marine cyanobacterium *Trichodesmium* spp., which was considered to be related to mycosporine-like amino acids components[38], but no fluorescent data were given. However, no direct evidence is given to date that pure cultures of heterotrophic bacteria, phytoplankton or zooplankton can produce refractory FDOM similar to that found in the deep ocean, and hence the origin of FDOM in the deep ocean remains largely unknown.

This study uses a combination of spectroscopic (ultraviolet–Vis absorbance, fluorescence and nuclear magnetic resonance (NMR)) and spectrometric (ultrahigh-resolution MS) techniques, as well as apparent fluorescence QYs and photochemical degradation experiments to investigate the optical properties and molecular composition of picocyanobacteria-derived DOM, which can be considered as a conceivable source of marine autochthonous CDOM (FDOM), that is also efficiently transported to the deep sea via picocyanobacteria cells packed in faecal pellets of zooplankton[8,9]. Results clearly show that lysed cells of *Synechoccocus* and *Prochlorococcus* pure cultures release FDOM that shows similar fluorescence pattern to deep-seawater samples. This 'humic-like' appearance of picocyanobacteria-derived FDOM is also comparable to deep-sea FDOM, by means of photo-degradation behaviour, apparent fluorescent QY measurements and its persistence during a 3-month dark incubation experiment. We conclude that picocyanobacteria-derived FDOM share very similar optical properties with marine FDOM.

## Results

**Optical properties of picocyanobacteria-derived CDOM.** Here we show that marine picocyanobacterial cultures grown in a medium of low-background CDOM (Supplementary Fig. 1) released high levels of FDOM (Fig. 1). EEM fluorescence and ultraviolet–Vis absorbance spectra of solid-phase-extracted DOM (SPE-DOM) from picocyanobacteria *Synechococcus* (Fig. 1a) and *Prochlorococcus* (Fig. 1b) cultures, and deep-ocean SPE-DOM (Fig. 1c, Sargasso Sea, 4,530 m depth) showed highly similar patterns, including the large Stokes shift of the previously defined humic-like fluorescence peak at a maximum excitation/emission of 250/450 nm (Fig. 2). In contrast, a heterotrophic bacterial culture (*Ruegeria pomeroyi* DSS-3) did not show these very specific fluorescence and absorbance signals (Figs 1d and 2). This typical humic-like peak has also been shown to increase with depth and exhibit relatively stable behaviour below 1,000–1,400 m in the global ocean[10,14,27].

Statistical EEM-PARAFAC analysis of the fluorescence data further strengthened the perceived relationship between picocyanobacteria-derived and marine FDOM. EEMs collected from SPE-DOM of a *Synechococcus* culture during a 15-day growth experiment showed variations that fitted best to a three-component PARAFAC model (Fig. 3 and Supplementary Fig. 2).

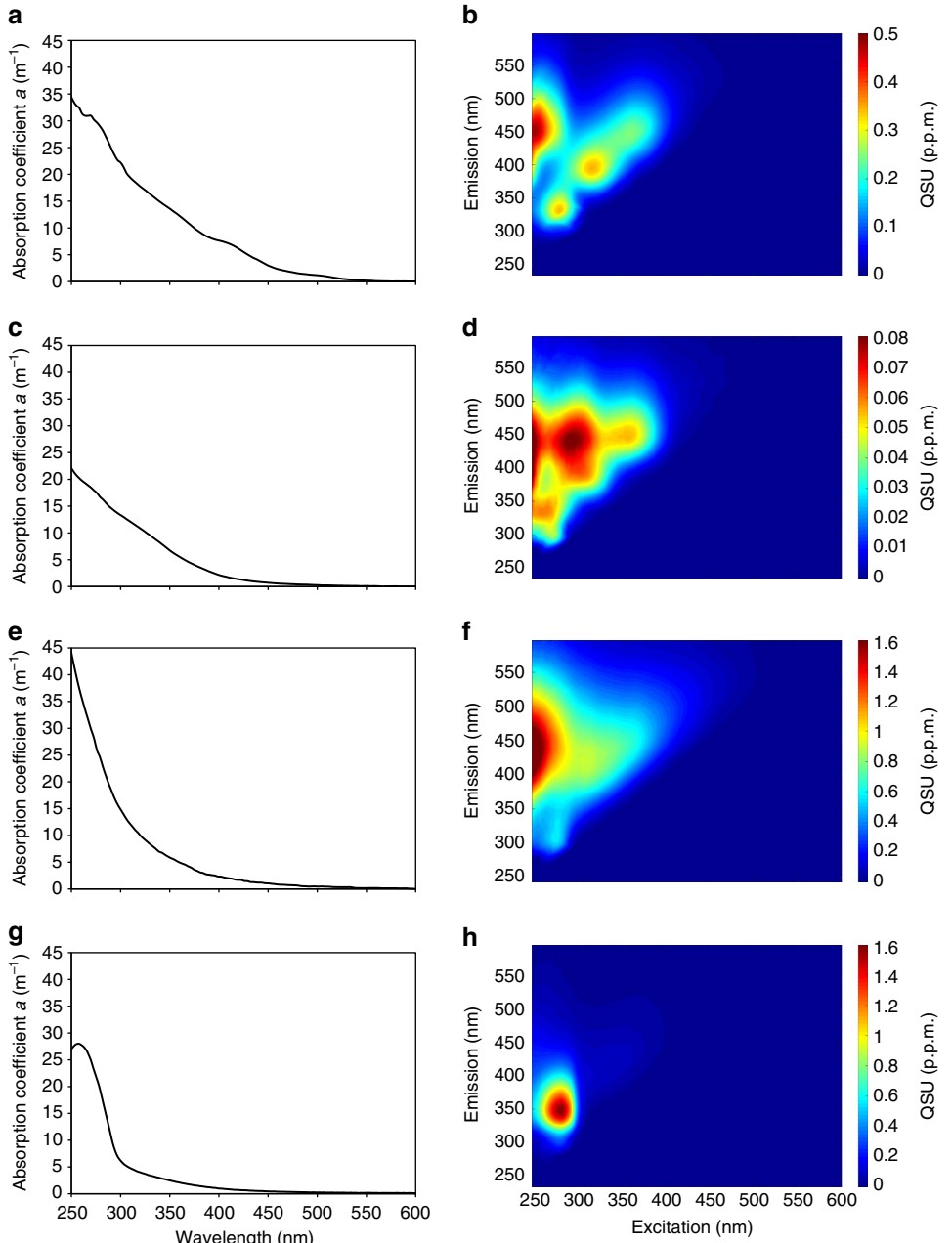

**Figure 1 | Optical properties of picocyanobacteria, marine and heterotrophic bacteria DOM.** Ultraviolet–Vis absorption and EEM fluorescence spectra of (**a,b**) *Synechococcus*-derived SPE-DOM, (**c,d**) *Prochlorococcus*-derived SPE-DOM, (**e,f**) SPE-DOM collected from the Sargasso Sea (BATS at 4,530 m depth) in August 2013 and (**g,h**) heterotrophic bacterium *R. pomeroyi*-derived SPE-DOM. Note: cell density was different in each culture and preclude a direct comparison of fluorescence intensity, and hence the given ultraviolet–Vis and EEM data are only intended to compare peak shapes and not intensities.

The three *Synechococcus* PARAFAC components (SYN1: (ex|em) 250 (350)|450 nm; SYN2: 250 (320)|400 nm; and SYN3: 270|320 nm) matched closely the three PARAFAC components obtained after modelling 24 marine SPE-DOM samples collected along a depth profile between surface and 4,530 m at the Bermuda Atlantic Time Series Station (BATS1: (ex|em) 250 (340)|460 nm; BATS2: 250 (310)|400 nm; and BATS3: 270|320 nm; Fig. 3 and Supplementary Fig. 3). Compared with a previous study of global oceanic FDOM[19], SYN1 showed an analogous positioning to a humic-like component (C1) with increasing fluorescent intensities down to a depth of about 1,000 m and then it remained constant in the deep ocean. SYN2 was comparable to component C4 of that same study[19], indicating the close similarity of our PARAFAC components when compared with global data sets.

SYN3 was the typical fluorescent signals often associated with proteins[39]. To further strengthen the similarity of SYN1–3 with marine FDOM, we compared these components with PARAFAC models published in the online spectral database Openfluor[40], and several matches within marine data sets were found for each component (Supplementary Fig. 4).

The intensity of all PARAFAC components increased with time during the growth experiments of cultured *Synechococcus* (Fig. 4). The sharp increase of these components during 15 days, along with the decrease in cell abundance of *Synechococcus*, indicated that the fluorescent chromophores were mainly derived from cell lysis products and not directly associated with *Synechococcus* cell exudates. Virus infection caused rapid and marked cell lysis leading to a large increase of FDOM with similar

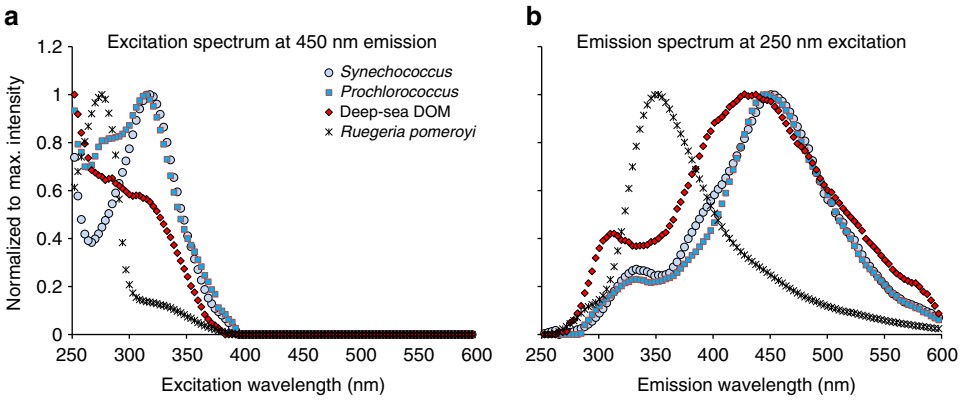

**Figure 2 | Excitation and emission spectra of *Synechococcus*, marine and heterotrophic bacteria FDOM at the humic-like fluorescence maximum.** Excitation and emission spectra, normalized to highest intensities of SPE-DOM, in pure water at maximum humic-like fluorescence at (**a**) excitation: 250 nm and (**b**) emission: 450 nm.

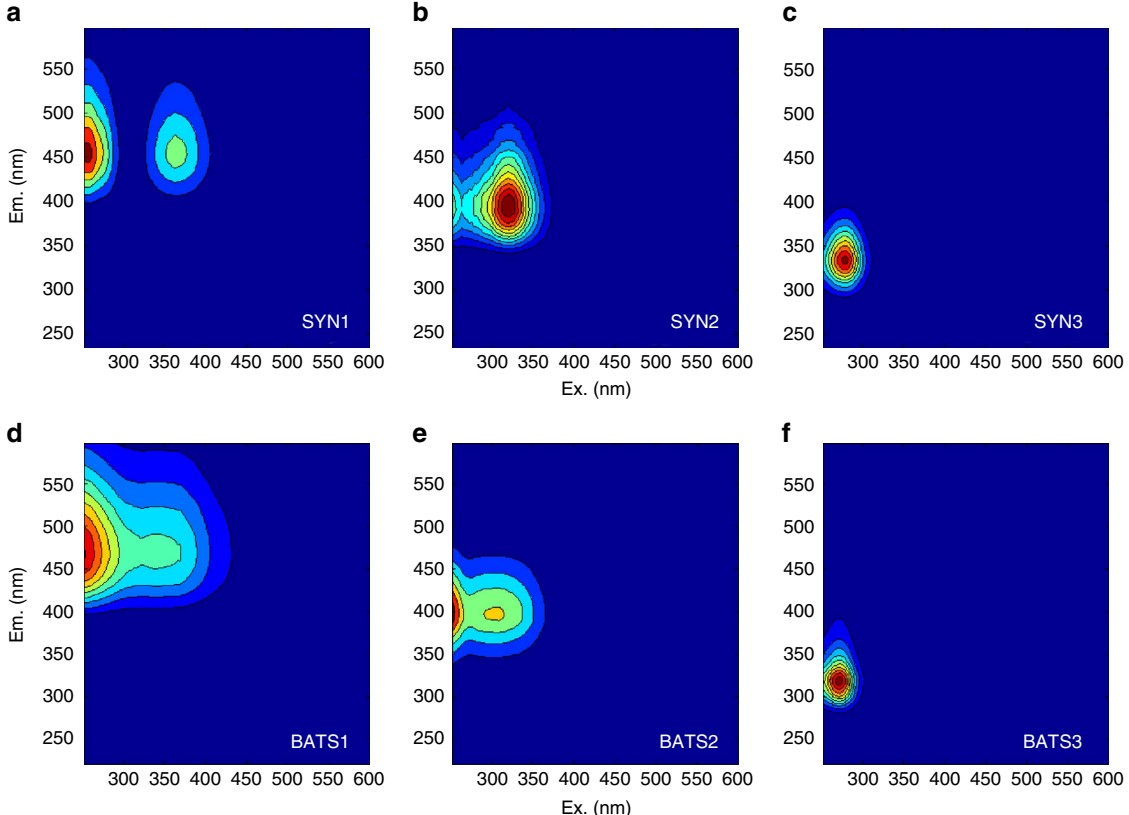

**Figure 3 | EEM-PARAFAC components of *Synechococcus* and marine FDOM.** EEM-PARAFAC three-component model of SPE-DOM of (**a**–**c**) a *Synechococcus* culture (CB0101) collected from a 15 days' growth experiment (SYN1–3) and (**d**–**f**) a marine SPE-DOM (BATS1–3) collected at every 200 m depth between surface and 4,530 m at the BATS.

fluorescent signals (Supplementary Fig. 5). The relative changes in the EEM fluorescence data indicated that the *Synechococcus* FDOM was not simply produced over time, but underwent further transformation during the 15 days' growth experiment, but maintained the humic-like characteristics.

**Persistence of picocyanobacterial FDOM.** To evaluate the bioavailability of picocyanobacterial FDOM, we added *Synechococcus* FDOM produced by viral lysis to coastal sea water and kept it in the dark for 3 months (Supplementary Fig. 6). All three *Synechococcus* FDOM PARAFAC components were found at much higher levels when compared to the seawater blank sample. The protein-like component SYN3 increased during the

first 3 days along with growth of heterotrophic bacteria, followed by a sharp decrease of that signal after day 3, presumably indicative of the depletion of readily available labile DOM. Component SYN1 showed a slight increase in intensity in the first 10 days, followed by a near-continual slow intensity decrease with an overall loss of about 12%. Component SYN2 showed an increase over the first 3 months by 23%, following a slight initial drop during the first day and near-constant intensities from days 30 to 90.

**Apparent fluorescent QYs.** Apparent fluorescent QYs are important indicators of the relationships between absorbance and fluorescence. In the case of an extensive similarity between

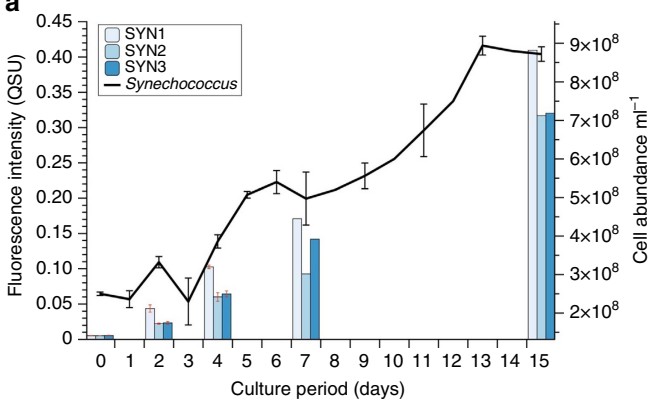

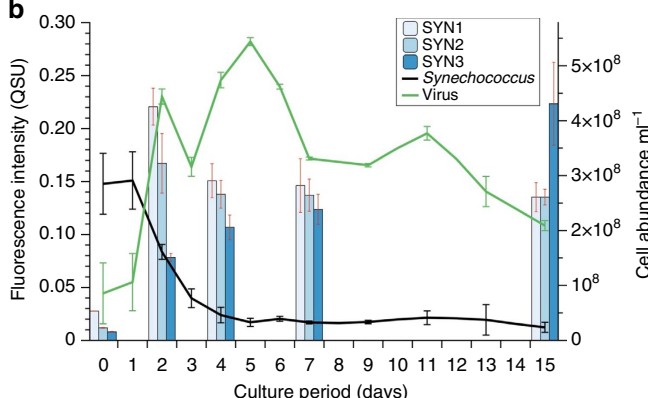

**Figure 4 | *Synechococcus* growth experiment with and without viral-induced lysis.** Fluorescence intensities (SYN1, SYN2 and SYN3) derived from the EEM-PARAFAC model of the cultured *Synechococcus* (CB0101) without (**a**) and with (**b**) virus during a 15 days' growth experiment. Error bars are expressed as s.d.

picocyanobacteria and marine FDOM, their respective apparent fluorescence QY should also match reasonably well. Apparent fluorescent QYs were determined for *Prochlorococcus*, *Synechococcus*, *Roseobacter*, marine-surface, marine-deep and terrestrially derived FDOM samples (Supplementary Fig. 7). The apparent fluorescence QYs of the deep-sea DOM sample (Sargasso Sea SPE-DOM collected at 4,530 m depth) showed a distinct local maximum at 390 nm, which was relatively close to the local maximum at 375 nm of the *Prochlorococcus* and *Synechococcus* SPE-DOM samples. The QY of the deep-sea DOM fell between the two strains of picocyanobacteria at most wavelengths, except for slightly higher QY of the deep-sea DOM sample above 410 nm. The QY is an important indication about the properties of the underlying chromophores. The QYs of picocyanobacteria and deep-sea DOM were in a similar range, which is a prerequisite to claim any similarities of the chromophores. The terrestrially derived Suwannee River natural organic matter (SRNOM) sample as well as the heterotrophic bacterium (*R. pomeroyi* DSS3) SPE-DOM samples showed much smaller QY when compared to deep-ocean or picocyanobacterial DOM samples. Further, only a small local maximum of the bacterial SPE-DOM QY was apparent at a much shorter wavelength of 285 nm, possibly corresponding to chromophores present in the protein-like fluorescence signal. The SRNOM QY did not show any pronounced local maximum, and QYs remained low throughout the recorded wavelength range, which is a typical feature of terrestrially derived DOM. Thus far, limited QY measurements were made on marine DOM, and a more substantial

data set would be needed to draw any further conclusions from these wavelength-dependent QY measurements.

**Photodegradation experiments**. Reproducible photodegradation experiments[10,41] undertaken on the deep-sea SPE-DOM (Sargasso Sea 4,530 m), SRNOM and three different strains of *Synechococcus* SPE-DOM (Supplementary Fig. 8) showed that the photodegradation of the *Synechococcus* SPE-DOM samples behaved in a similar manner to the deep-sea SPE-DOM, whereas the maximum fluorescence loss of SRNOM occurred at longer emission wavelengths. The highly similar photodegradation behaviour between marine and *Synechococcus* FDOM further strengthened the hypothesis that picocyanobacteria might significantly contribute to total marine FDOM.

**Influence of bacteria on *Synechococcus*-derived FDOM**. The production of 'protein-like' PARAFAC component SYN3 was closely related to heterotrophic bacterial activity, indicating an origin of bacterial metabolism. Similar fluorescent signals were detected in the single-strain culture of *R. pomeroyi* DSS-3. However, humic-like components SYN1 and SYN2 were more stable during bacterial consumption and degradation. Both humic-like components were at low and steady levels in the seawater control (Supplementary Fig. 6). Slight changes may occur due to modification of *Synechococcus* FDOM caused by bacterial activity such as extracellular enzyme degradation[42]. *Synechococcus*-associated bacteria were demonstrated to be a symbiotic system attached within cells in some cases[43], and bacteria were detected in the monoculture of *Synechococcus* used in the 15 days' incubation experiment, which was initially purified with plating procedures[44]. We tested the strain with purity test broth[45,46], and our field strain CB0101, isolated from the Chesapeake Bay[47], was demonstrated to be a non-axenic monoculture. However, in this study bacteria present in the *Synechococcus* cultures were at a low ratio (6% of cell abundance) and the bacterial abundance was consistent during the growth or viral-lysis processes of *Synechococcus* cells. FDOM accumulation during growth, especially when cells were lysed by viruses, indicated that cell detritus of *Synechococcus* contributed mainly to this FDOM. To further evaluate whether bacteria associated with our CB0101 culture contributed to the FDOM signals, we isolated these bacteria from the purity test broth and evaluated their optical properties (Supplementary Fig. 9). We also created an axenic *Synechococcus* strain culture WH7803 to leave no doubt that the observed FDOM signal originated from the picocyanobacteria themselves and not at all derived from bacteria. The comparison between the original *Synechococcus* sample (non-axenic, Fig. 1) and the axenic culture (Supplementary Fig. 10) showed a very close match, with the exception that the protein-like fluorescence was not apparent in the axenic culture. The isolated bacteria culture from *Synechococcus* strain CB0101 showed only very weak fluorescence and no detectable fluorescence at emission 450 nm (Supplementary Fig. 9).

**Chemical composition of *Synechococcus*-derived DOM**. Using ultrahigh-resolution mass spectrometry (Fourier transform ion cyclotron resonance MS), analysis of *Synechococcus* SPE-DOM revealed a large diversity of organic compounds originating from *Synechococcus* cells (Fig. 5). Only a rather restricted fraction of the assigned molecular formulas of carbon, hydrogen and oxygen (CHO) compounds, and a very diverse group of nitrogen-containing (CHNO) compounds were sufficiently hydrogen-deficient to be plausible candidate molecules causing the observed fluorescence signatures. Here the most intense CHNO ions contained two nitrogen atoms. To

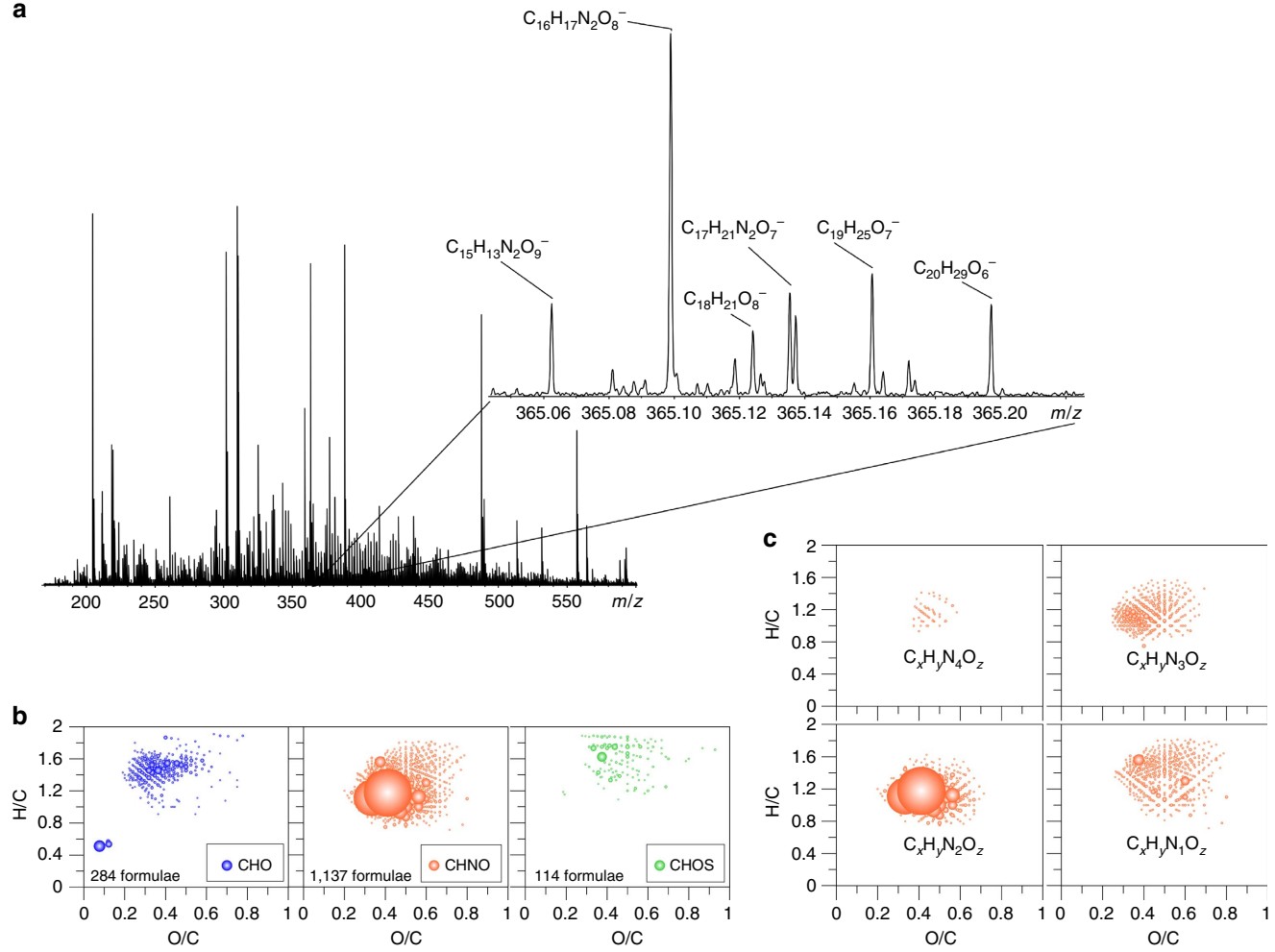

**Figure 5 | Ultrahigh resolution mass spectrum of *Synechococcus* DOM.** (**a**) Fourier transform ion cyclotron resonance mass spectrum of *Synechococcus* SPE-DOM, (**b**) van Krevelen diagram of all assigned molecular formulae of *Synechococcus* (CB0101) SPE-DOM and (**c**) van Krevelen diagrams of the distribution of CHNO formulae. Note: size of bubbles represents relative abundance.

further evaluate the possibility that organic nitrogen compounds were responsible for the observed optical properties, 800 MHz [1]H NMR spectra were acquired. [1]H-NMR resonances of the H–N bond (Supplementary Fig. 11 and Supplementary Table 1) were abundant and indicative of pyrrolic ring structures, and suggested the presence of chromophores such as phycobilins or metabolites with intact pyrrolic units. The parent chromophore phycocyanobilin showed a maximum absorbance around 650 nm (ref. 48) and hence a fluorescent signal at even longer wavelengths[49]. However, phycobilins are very sensitive to light, and it is likely that the observed fluorophores in this study were metabolites that were rather resistant to microbial degradation and underwent much slower further photo-degradation. Evidence for this hypothesis was provided by a photo-degradation experiment, in which phycocyanobilin (Frontier Scientific) was dissolved in water and exposed to 22 h simulated sunlight, similar to the described photo-degradation experiment of *Synechococcus* SPE-DOM. Here the pyrrolic chromophore phycocyanobilin underwent rapid photo-degradation as evident by the fast decrease in absorbance at longer wavelengths (Supplementary Fig. 12). The photo-production fluorophores with very high apparent QY underlined the highly fluorescent nature of photo-metabolites of phycocyanobilin (Supplementary Fig. 13). These intense produced fluorophores showed fluorescence that was not

very different to marine and picocyanobacteria FDOM, and showed only a peak shift of 15 nm from the marine FDOM excitation wavelengths. The large difference in apparent fluorescent QYs suggested that structural differences still exist between photochemically produced phycocyanobilin metabolites and the fluorophores found in our picocyanobacteria cultures, but this result was not surprising given the very different chemistry at play during photolysis and chemical processes taking place after cell lysis. However, the fluorescence similarity of photo-produced FDOM from phycocyanobilin metabolites and picocyanobacteria FDOM is further evidence that a suite of phycocyanobilin metabolites might have caused the FDOM fluorescence signatures observed in picocyanobacteria culture. The abundant and diverse suite of organic nitrogen compounds containing mostly two nitrogen atoms (Supplementary Table 2) also pointed towards organic nitrogen-containing fluorophores, although at this point the structure of a direct fluorescent metabolite of phycocyanobilin could not be determined.

**Possible fluorescent metabolites of *Synechococcus*-derived DOM.** There are multiple biological sources of pyrroles in the environment, including many biologically active natural products and porphyrins such as chlorophylls. Degradation products of these pyrrolic materials may also contribute to global FDOM

production at the surface, however most of them, including chlorophylls, are not readily water-soluble, but water-soluble stable metabolites may be formed. For example, chlorin-P is known to be a major water-soluble metabolite of chlorophyll, but it does not fluoresce in the humic-like fluorescence region. Further, humic-like fluorescence have never been associated with chlorophyll metabolites in general. A complex suite of tetrapyrrolic phytochromes have been described in the literature[50], but no evidence have been given to date that porphyrin-related degradation products show similar optical properties to marine CDOM.

The phycobilin pigments in *Synechococcus* are readily water-soluble and have been detected in the surface and mixed layer of the ocean, but also at depth. However, the tetrapyrrolic phycobilins cannot explain marine humic-like fluorescence either, because of the distinct fluorescence at much higher wavelengths. In this study, we make the case that fluorescent metabolites of phycobilins may explain marine CDOM, but these metabolites cannot maintain the tetrapyrrolic structure to be able to show similar optical properties to marine CDOM.

**Picoycyanobacteria FDOM export to the deep ocean.** Marine picocyanobacteria are globally distributed and even dominant in oligotrophic areas where phytoplankton primary production is relatively low and hence have the potential to be a major source of marine FDOM. Picocyanobacterial pigments (for example, phycoerythrin) also have been shown to be more readily exported down to deep ocean layers than chlorophyll or other phytoplankton pigments, according to sediment trap records[9]. In our laboratory culture experiment, the Fmax production rate of humic-like EEM-PARAFAC component (SYN1, Fig. 4) was $3.71 \pm 1.84 \times 10^{-12}$ $QSU_m$ per cell. It should be noted here that the fluorescence was converted into equivalent mass of quinine sulfate ($QSU_m$, see Methods section). According to sediment trap records from the Costa Rica upwelling dome, the total *Synechococcus* export to depth ranged between 0.04 and 1.06% of the standing stock of *Synechococcus*[9]. The annual mean global abundance of *Synechococcus* was also previously estimated based on >35,000 observations to be $7.0 \pm 0.3 \times 10^{26}$ cells[51]. By using the percentage range (0.04–1.06%) of exported *Synechococcus* cells to depth mentioned above, we can calculate a global annual humic-like fluorescence export ranging between $1.04 \times 10^{12}$ and $2.75 \times 10^{13}$ $QSU_m$. This estimate is grossly underestimating the whole contribution from all picocyanobacteria to the biological pump, because *Prochlorococcus* was not included in this calculation, but underlines that *Synechococcus* alone has the potential to substantially contribute FDOM to the deep ocean. A previous study also investigated the contribution of pigment degradation products to the marine CDOM pool and suggested a porphyrin source of unknown origin[52].

We conclude that picocyanobacteria-derived FDOM is similar to Sargasso Sea deep-sea FDOM in terms of optical properties, apparent fluorescence QYs and photo-degradation behaviour. The humic-like component, which is a common feature of marine FDOM, was confirmed being present in picocyanobacterial FDOM at high levels. Metabolites of the chromophore phycocyanobilin might be contributing to this specific *Synechococcus* fluorescence. *Prochlorococcus* lacks the genes to synthesize most phytochromes, but it still has the ability to synthesize phycoerythrin and its chromophores phycoerythrobilin or phycourobilin[53], which are phycobilins.

We demonstrated in this study that picocyanobacteria are well suited to be one important source of organic fluorophores in the World's Ocean, but do not claim that it is the only one. Picocyanobacterial nitrogen-rich products associated with FDOM might be of special interest regarding nitrogen transport and cycling in open ocean environments and our findings are likely to be significant in a changing climate due to the fact that picocyanobacterial abundance is expected to increase with increasing temperature[51].

## Methods

**Bacterial cultures.** Marine *Synechococcus* strains CB0101, WH7803, *Prochlorococcus* strain MIT9313 and a heterotrophic marine bacterium *R. pomeroyi* strain DSS-3 were used in this study. *Synechococcus* CB0101 were grown in the modified SN medium[54] without EDTA and vitamin $B_{12}$ (referred to as medium SN15 thereafter) at 25 °C under constant cool white light (20 to 30 µE m$^{-2}$ s$^{-1}$). *Prochlorococcus* MIT9313 culture was cultivated in the Hong Kong Port Shelter seawater-based Pro99 medium[55] at 21 °C under constant cool white light (30 µE m$^{-2}$ s$^{-1}$). *R. pomeroyi* DSS-3 was kindly provided by Dr Mary Ann Moran at the University of Georgia. Strain DSS-3 was grown in the YTSS medium (4 g l$^{-1}$ yeast extract, 2.5 g l$^{-1}$ tryptone and 20 g l$^{-1}$ crystal sea salt) at 28 °C. Other cultures may be more representative of open ocean environments (for example, SAR11), but are very difficult to culture. However, the heterotrophic bacteria associated with *Synechococcus* CB0101 were also cultured for comparison.

All picocyanobacteria strains (CB0101, WH7803 and MIT9313) were tested in Marine Purity Broth[56], ProAC[57], ProMM[58] and ASW solid medium with yeast extract[45], respectively. CB0101-associated bacterial assemblages were isolated from the ASW solid medium. All colonies were pellet washed with 5 ml sterilized SN15 medium and inoculated into SN15 medium with cell abundance adjusted to $10^7$ ml$^{-1}$. CB0101-associated bacteria were cultured under the same condition as CB0101.

**DOM collection.** *Synechococcus* and *R. pomeroyi* cultures were filtered through 0.7 µm GF/F (Whatman) glass fiber filters, acidified to pH 2 and solid-phase extracted (SPE) for DOM using the method described previously[59]. At incubation day 7, *Prochlorococcus* MIT9313 culture was filtered through a 0.2 µm filter (Acrodisc Syringe Filters with Supor Membrane PN4612) and then through a 0.02 µm filter membrane (Whatman Anodisc inorganic filter membrane WHA68096002). This filtered culture sample was then extracted as described above, and 1 ml of the methanolic extract was dried and re-dissolved in 10 ml pure water before all optical property analyses. The used SPE method is well suited to evaluate the CDOM pool, because of its very good recovery using the styrene-divinylbenzene polymer that has been modified with a proprietary nonpolar surface (Agilent Bond Elut PPL). No significant differences between extracted and non-extracted samples were found, beside the much better signal to noise ratio of concentrated SPE-DOM.

Additional 20 ml samples were also collected for the fluorescent spectroscopic analyses from the GF/F-filtered replicate *Synechococcus* CB0101 cultures and from the parallel cultures that were inoculated with the viral strain P1 at incubation day 0, 2, 4, 7 and 15, respectively. The 15 days' incubation experiment was conducted under the same growth condition with additional gentle air bubbling to maintain a culture volume of 500 ml.

*Synechococcus* DOM of strain CB0101 was collected at day 5 of the culture and inoculated with viral stain P1, which caused a rapid and natural cell lysis of strain CB0101. The seawater samples were collected in close proximity to Ocean City, Maryland and filtered through 0.8 µm filters. The filtered seawater samples were kept in the dark for 5 days before the addition of *Synechococcus* DOM to reach a steady DOM background. A volume of 1 l of the filtered culture and SN medium was then added into the 10 l costal seawater samples, and triplicate treatments were incubated at room temperature for 90 days in the dark. A volume of 1 l water samples were then collected at days 0, 1, 3, 5, 10, 30, 60 and 90, subjected to SPE and 1 ml of extract was dried and re-dissolved in pure water for optical property analyses (ultraviolet–Vis and EEMs).

A volume of 1 ml samples were taken during both the 15 days' culture and the 90 days' incubation experiments. Cell abundance of *Synechococcus*, bacteria and viruses were analysed on an Epics Altra II flow cytometer (Beckman Coulter, USA) with a 306C-5 argon laser (Coherent, USA). The enumeration was performed following original methods of Brussaard[60], Marie *et al.*[61] and Jiao *et al.*[62] Sample preparation and laser settings followed the protocol described by Liang *et al.*[63]

**Marine DOM samples.** The 10 l open ocean samples were collected between surface and 4,530 m at 200 m depth intervals at the BATS in July 2013 aboard the RV Atlantic Explorer and SPE using the Agilent Bond Elut PPL resin[59]. The 10 l seawater samples were filtered through GFF filters (Whatman), acidified to pH 2 using formic acid (Sigma-Aldrich, 98%), instead of HCl to avoid chloride adducts formation in the electrospray and then extracted using 1 g Agilent Bond Elut PPL cartridges. After extraction, the cartridges were washed with acidified pure water, dried and eluted off with 10 ml methanol (Chromasolv liquid chromatography–MS, Sigma-Aldrich).

**Excitation emission matrix fluorescence.** The EEM fluorescence spectra from the open-ocean, autotrophic and heterotrophic bacteria were recorded on the dried and re-dissolved (water) SPE-DOM. The 20 ml samples from the 15 days' *Synechococcus* growth experiment were filtered (GFF Whatman) and directly measured. We also conducted an additional growth experiment where 100 ml sample was SPE to be able to compare the effect of solid-phase extraction and no differences were found between extracted and direct samples after correction for concentration. The absorbance of all samples were determined simultaneously with the EEMs using the Jobin Horiba Aqualog fluorometer and the raw absorbance was converted to apparent absorption coefficients[64]. The emission ranged between 211 and 617 nm and excitation between 230 and 600 nm. All fluorescence spectra were scatter-corrected, inner filter corrected using the simultaneously measured ultraviolet–Vis spectrum and normalized to a $1\,mg\,l^{-1}$ quinine sulfate (QS) standard (Starna reference material RM-QS00, $1.28 \times 10^{-6}\,mol\,l^{-1}$); hence we expressed all fluorescence intensities in QSU (p.p.m.). We also used the QS equivalent mass ($QSU_m$) for calculating standing stocks of fluorescence expressed in milligrams of QS. Individual PARAFAC models were built using the marine SPE-DOM data set as well as EEMs derived during a 15 days' growing experiment of the *Synechococcus* strain CB0101. The PARAFAC modelling was based on the MATLAB scripts used in the drEEM toolbox[17]. The marine DOM three-component PARAFAC model was split-half validated (Supplementary Fig. 2) with a core consistency of 89.9, and 99.2% of the data was explained. A three-component PARAFAC model of the *Synechococcus* culture experiment was also created and split-half validated (Supplementary Fig. 3) with a very high core consistency of 96.3, and 96.6% of the data was explained. Four-, five- and six-component models were also evaluated but did not yield feasible results.

**Photo-degradation experiments.** A custom-built irradiation system with a solar simulator was used to irradiate samples[41]. A volume of 1 ml of the previously dried (removal of methanol) SPE-DOM was re-dissolved in pure water and added to an equilibrator vial containing a stir bar to ensure proper aeration and prevent oxygen starvation during irradiation. A pH probe, connected to a dual pump syringe system filled with sodium hydroxide and HCl, respectively, ensured constant pH during irradiation through microinjections of acid or base depending on pH drift. The sample was pumped continuously through the flow cell of the Aqualog and through a custom-built irradiation cell that was exposed to the collimated beam of the solar simulator. Continuous circulation of the sample ensured consistent exposure. The irradiation cell was kept on cooling blocks regulated by a chilling system to keep a constant temperature of 25 °C. Before irradiation, lamp intensity was checked to ensure that the entire irradiation cell received $100 \pm 3\%$ of the light intensity equivalent to the radiation of the sun at the Earth's surface and at 45° north, midsummer, at 12:00. Thus, temperature, pH, light intensity and sample exposure were kept constant throughout the experiment to ensure any changes in optical properties were due solely to irradiation. This system allowed for very reproducible photodegradation experiments and enabled a direct comparison between different sources of DOM.

**Apparent fluorescent QYs.** Apparent fluorescent QYs were measured on all SPE-DOM samples and the concentrations were adjusted to achieve an absorbance (A) at 300 nm of 0.4 units. The apparent QYs were calculated according to previously published procedures[65,66] and using QS as a QY standard[67]. The Horiba Aqualog was ideal to compute accurate apparent fluorescence QYs because of the simultaneous measurements of absorbance and fluorescence, and the accurate inner filtering correction.

**Ultrahigh-resolution MS.** A non-target ultrahigh-resolution MS approach was used to characterize the complex DOM released by *Synechococcus* strain CB0101. All SPE-DOM samples were analysed at the Helmholtz Center for Environmental Health, Munich, Germany using a 12 T Bruker Solarix Fourier transform ion cyclotron resonance mass spectrometer interfaced with negative-mode electrospray ionization. This soft ionization in combination with ultrahigh-resolution MS allowed to accurately assign exact molecular formulae to the observed singly charged $m/z$ molecular ions. Mass accuracy exceeded $<0.2$ p.p.m. at a mass resolution of $>500,000$ at $m/z$ 400. Methanolic samples were infused at a flow rate of $120\,\mu l\,h^{-1}$, electrospray voltage was set to 3,600 V and 500 scans were averaged. All samples were analysed in triplicates and only $m/z$ ions consistently present in all spectra at comparable relative abundances were considered. Results were plotted using van Krevelen diagrams[68], in which the hydrogen to carbon ratio (H/C) was plotted against the oxygen to carbon ratio (O/C) of assigned molecular formulas.

**¹H nuclear magnetic resonance mass spectrometry.** ¹H NMR-detected spectra of methanolic *Synechococcus* SPE-DOM extract were acquired with a Bruker Avance NMR spectrometer at 800.13 MHz ($B_0 = 18.7\,T$) at 283 K from $\sim 0.5\,mg$ of solid obtained by evaporation of original methanol-$h_4$ solution, dissolved in 69 mg $CD_3OD$ (Merck, 99.95% $^2H$) solution with a 5 mm z-gradient $^1H|^{13}C|^{15}N|^{31}P$ QCI cryogenic probe ($^1H$ 90° excitation pulse: 11.2 μs) in a sealed 2.0 mm Bruker MATCH tube. One-dimensional $^1H$ NMR spectra were recorded with a spin-echo sequence (10 μs delay) to allow for high-Q probe ringdown, and classical

pre-saturation to attenuate residual water present 'noesypr1d', 5,377 scans (5 s acquisition time, 5 s relaxation delay, 1 ms mixing time; 1 Hz exponential line broadening). A phase-sensitive, gradient-enhanced total correlation spectroscopy (TOCSY) NMR spectrum with solvent suppression (dipsi2etgpsi19) was acquired for an acquisition time of 1.7 s, a mixing time of 70 ms and a relaxation delay of 0.8 s (128 scans|1,222 increments at 9,615.4 Hz|12.0 p.p.m. bandwidth).

**Data availability.** The data reported in this article are partially available in the Supplementary Material, and metadata can be acquired directly from the corresponding author M.G. on request.

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

## Acknowledgements

This research was funded by the University of Maryland Center for Environmental Science, Institute of Marine and Environmental Technology (IMET) Seed Grant (awarded to F.C. and M.G. in 2015). This is contribution 5314 of the University of Maryland Center for Environmental Science. This project was partially supported by the Maryland Sea Grant REU programme in summer 2015. This publication is partially supported by grants to N.J. in from the Ministry of Science and Technology (2013CB955700), the Natural Science Foundation of China (91428308) and State Ocean Administration (GASI-03-01-02-05), to L.X. from the Ministry of Science and Technology (2016YFA0601103), and to Q.Z. from the Research Grants Council of the Hong Kong Special Administrative Region, China (16103414) and the Natural Science Foundation of China (41476147).

## Author contributions

M.G. and F.C. conceived the study and designed the experiments; Z.Z. performed experiments and data analyses; J.L., S.T. and H.I. undertook optical properties analyses and photochemical degradation experiments; N.H. undertook NMR analysis and P.S.-K. assisted in ultrahigh-resolution Fourier transform ion cyclotron resonance mass spectrometer analysis; X.F. and Q.Z. provided the lysates of *Prochlorococcus* cultures; N.J. provided cell counts using flow cytometry. M.G., Z.Z. and F.C. wrote the manuscript with feedback from all authors.

## Additional information

**Competing interests:** The authors declare no competing financial interests.

**Publisher's note**: 

