## [Peer Review File · Nature Communications]

Reviewers' Comments:

Reviewer #1 (Remarks to the Author)

Review of NCOMMS-16-09020-T. Gonsoir et al. "Picocyanobacteria and Deep-Ocean Fluorescent Dissolved Organic matter share the same Optical Properties",

This contribution is focused on linking the origin of fluorescent dissolved organic matter in the ocean to *Synechococcus*. As the authors explain there is a wide distribution of FDOM with fluorescence at visible wavelengths (so called humic-like fluorescence) in the ocean and evidence for its net accumulation in the deep ocean. The actual mechanisms involved in its formation and the organisms responsible is still currently unknown although there is evidence for direct exudation by phytoplankton and also production associated with the processing of DOM by heterotrophic bacteria. Insight on the actual chemical compounds responsible is still lacking and this contribution provides some valuable evidence in this light.

The contribution is broken into several parts. First the authors present results from cultures with *Synechococcus*. The FDOM accumulated in the cultures has similar fluorescence properties to marine DOM although this comparison could be strengthened to be more convincing (see point 1). Next they compare the dynamics of FDOM accumulation and development in *Synechococcus* cell abundance. Here I am not convinced that there is a clear direct link to *Synechococcus* (see point 2). How can the processing of *Synechococcus* released DOM, by heterotrophic bacteria be ruled out here? Next they provide additional optical data in the form of apparent quantum yields and effects of photochemical exposure. These data, in my opinion, although of good quality and interesting in their own right, do not support the case they are trying to make. (See point 3). Finally the authors present the result of the chemical characterization of the DOM accumulated in the cultures and this leads them to conclude a significant presence of pyrole structures. These likely have their origins in plankton pigments and can be expected to be a ubiquitous source in the ocean. They carry out measurements on commercially available phycocyanobilin and find that when photodegraded it has an optical signature similar to that of humic marine DOM. This is a substantial finding and can easily be further cemented by including the AQY spectra (See point 4). In its current form I do not find the manuscript suitable for publication. Yes I am convinced that phytoplankton pigments are the ultimate precursor material for FDOM but is there a reason to expect that the FDOM precursor material released by picocyanobacteria is specific to them and not released by other organism (phytoplankton at least)? It is clear that some subsequent extracellular abiotic and biotic processing of this material is required. The evidence presented does not provide a convincing argument for why picocyanobacteria should dominate the production, rather than this being a more widespread source.

1.

Lines 110-116. It is difficult to follow this comparison with the way the fluorescence data are currently plotted (contour plots). It would be easier if the authors selected a "slice". This could be a specific emission spectrum or even a diagonal which captures the main features (e.g. synchronous scan). Space is at a premium, but the contour plots could be made smaller for example as an inset to a 2d plot of the slice. Finally the comparison to bacteria is also difficult as there is such a large peak at Em350. I wonder if the scale is zoomed in maybe the humic-like signal with large Stokes Shifts would appear.

Line 121-133. This comparison is also difficult to follow. Again a 2D plot like those in FigS2 and 3 would be better. I suggest replacing Fig 2 with a 2D plot of the excitation and emission loadings of each component compared. The match would then be clearer to the reader. I think it would also be apparent (as is from looking at the EEMs) that the emission spectra match very well, but the marine DOM excitation spectra are broader. Expand on your comparison to parafac components from earlier studies by including these components on plots (in sup info). These data should be

available on OpenFLuor.org

2.

Lines 134-141. It is stated that the increase in fluorescence in the non-virus amended *Synechococcus* culture (Fig 3) was linked to a decrease in cell abundance (FigS4). I can not see the decrease in cell abundance in the culture without virus. In addition it is difficult to evaluate the changes in bacteria numbers as the scale is too large. Bacteria numbers should be around a factor of 100 lower. In fact the greatest increase in FDOM is seen between day 4-7 and this is where it looks like there are greatest bacteria abundances. The FDOM data for the virus amended treatment is not shown (as in Fig3, only as a contour) but does this show the same pattern? Also it seems strange when comparing the contour plots in Fig S4 with the Fig3. For example day 7 has a low fluorescence (<0.25) but SYN1 in Fig3 is above 0.3. I think something is mixed.

Lines 134-141. It is stated that the increase in fluorescence in the non-virus amended *Synechococcus* culture (Fig 3) was linked to a decrease in cell abundance (FigS4). I can not see the decrease in cell abundance in the culture without virus. In addition it is difficult to evaluate the changes in bacteria numbers as the scale is too large. Bacteria numbers should be around a factor of 100 lower. In fact the greatest increase in FDOM is seen between day 4-7 and this is where it looks like there are greatest bacteria abundances. The FDOM data for the virus amended treatment is not shown (as in Fig3, only as a contour) but does this show the same pattern? Also it seems strange when comparing the contour plots in Fig S4 with the Fig3. For example day 7 has a low fluorescence (<0.25) but SYN1 in Fig3 is above 0.3. I think something is mixed.

3.

Line 159-171. The comparison of AQY is a great idea but it is difficult to follow the comparisons made. The AQY spectra in Fig4 appear very different from each other.

Line 175-178. The photodegradation results are very interesting but again it is difficult to follow the authors comments when looking at Fig S6. The clearest pattern apparent is that the FDOM loss is directly linked to the original FDOM signal. So the SRNOM has longer wavelength fluorescence and therefore also shows a loss of longer wavelength fluorescence.

4.

Line 179-205. This is a very interesting result. Can the authors include the AQY spectrum of phycocyanobilin before and after photodegradation. Is it also possible to include the other chemical measurements (MS, NMR). This would make for a very convincing argument.

Other points

Line 119. This sentence stands alone. What statistical analysis?

Line 142-147. I do not agree completely. The experiment has shown that the combination of *Synechococcus*, heterotrophic bacteria and virus can result in the production of FDOM signal, which is similar to marine DOM (in emission in particular).

Line 156-159. This statement is a little too bold. Dark yes but temperatures were higher and the overall substrate levels were also very different. Coastal water has higher background DOM levels.

Figure 1. It is unclear from the legend that the a & b are also SPE-DOM. Is explained in the text (line 110-111).

Reviewer #2 (Remarks to the Author)

I find this short yet complete manuscript to be very interesting and novel. It crosses some

interesting methods (fluorescence EEMs, NMR and FT-ICR-MS) to try to resolve the question of where deep sea FDOM comes from. Here the authors propose that a proportion of the deep sea FDOM is of picocyanobacterial origin (*Prochlorococcus* and *Synechococcus*). I find the paper to be of an acceptable quality for Nature Communications and have some small suggestions that will help to improve the clarity for the reader of the article.

I find the abstract to be clear, well written, appropriate and easy to follow.

Page 3, line 105: when the authors say pure, do they mean axenic or that they are non-axenic monocultures?

Line 106: where these measurements conducted on extracted or just filtered samples?

Line 112: I agree that the spectra are similar but I think that the authors should tone down the phrase 'striking congruence'. Also, what about surface water, does it look the same? Also, add in the letters for the deep ocean SPE-DOM.

115: did the bacterial culture have algal DOM as a carbon source?

118: yes, but this does not prove that all deep ocean FDOM is on picocyanobacteria origin.

167-169: yes, I agree that there are some nice parallels here between the quantum yields from the picocyanobacteria FDOM and the deep ocean sample. How do the authors explain the shifts? FDOM from other sources? Aging of the FDOM? Do the authors have any thoughts on how the spectra from other autotrophic sources might compare?

Page 15, line 14: did the authors acidify the samples for the fluorescence spectroscopy analysis? If so, this will have had a negative effect on the fluorescence measurements. See : Green and Blough (1994).

Page 15, line 18. Why was air bubbling not used in all of the cultures?

Line 29: coastal (check spelling)

Pg 16, line 52. Can the authors check the values for emission and excitation? Perhaps I am missing something, but it appears that the emission values were lower than the excitation values. I am wondering if the extraction process altered the DOM and hence the FDOM signals. Can the authors comment on this? Also as much work have been published on non-extracted FDOM, I think it is important that the authors clearly stated that their measurements were conducted on extracted samples on the graphs (it is noted in some of them but not all, eg. Fig. 4). This is a potential confusion point and it should be clearly dealt with in the figure legends and in the text.

Fig. 1. Please put the same y axis scale on the absorption graphs. Why are there horizontal lines on the figures? This should be explained in the legends.

Fig. S4. For ease of comparison, it would be nice to have the same y axis scaling on the cell abundance graphs.

Fig. S5. Please check the format of the upper figure, it looks a bit truncated. Maybe change to "lower" rather than "bottom" when referring to the panels in the graphs (here and elsewhere in the manuscript).

There is no use of statistics in this work. It would be good to have some idea of the replicability of the measurements however. plus how many replicate measurements were made for each analysis, etc.

I find the conclusion to be robust and to be based on the results presented.

I think the format of the journal names needs to be checked, some are in full whereas some are given as the abbreviated form. Please also check ref 1 of the methods section (SYnechococcus) Please check the format of Ref. 10 : the name should be corrected to: Romera-Castillo, C.

Green, S.A., Blough, N.V., 1994. Optical absorption and fluorescence properties of chromophoric dissolved organic matter in natural waters. *Limnol. Oceanogr.* 39, 1903-1916.

Reviewer #3 (Remarks to the Author)

The authors present a clear and compelling argument for cyanobacterial pigments contributing to the deep-ocean pool of fluorescent dissolved organic matter. This is a significant result as this work clearly establishes a pathway for an autochthonous source for CDOM components which has only been inferred in the past. This in turn ties into questions of the origin of deep-sea DOM and its lifetime, which are largely open to question. So I think this topic is very important. The authors use state of the art techniques in analysis and have combined observation and experiment in a logical way.

I would like to identify a couple areas in which the authors can strengthen their argument. First, it seems to me there's a bit of a jump from "tetrapyrrole" and "phycobilin" here. Phycobilins are not the only tetrapyrroles in the marine environment, and in fact it can easily be argued that they are far outnumbered by those found in chlorophylls and cytochromes (for example). The authors must have convinced themselves that the chemical structures they were seeing in the environment came from phycobilins and not from other chemical structures containing tetrapyrroles such as chlorophyll or cytochrome degradation products. Some clarification here, I think, is essential.

Also I think a bit more discussion is warranted on the differences (or lack thereof) between the spectra of terrestrial humic material and the cyanobacterial derived material. It is certain that there is some terrestrial humic material in the deep ocean, and the question is how much compared to the autochthonous fraction. The fluorescence signature of terrestrial humic material is highly conserved. Is the similarity between these substances of differing origin just a coincidence?

I have some minor quibbles regarding the use of GF/F filters to separate dissolved fractions from particles (DON'T DO THIS), and with the unquestioning use of SPE to concentrate dissolved organic matter. These processes are selective, and the impacts of this selection should be acknowledged.

Please see details below where we addressed all reviewers' comments. We
substantially revised the manuscript to explain our results in more detail as
requested by the reviewers. Our comments are in red.

**Reviewer #1:**

Review of NCOMMS-16-09020-T. Gonsior et al. "Picocyanobacteria and
Deep-Ocean Fluorescent Dissolved Organic matter share the same Optical
Properties": This contribution is focused on linking the origin of fluorescent
dissolved organic matter in the ocean to *Synechococcus*. As the authors
explain there is a wide distribution of FDOM with fluorescence at visible
wavelengths (so called humic-like fluorescence) in the ocean and evidence
for its net accumulation in the deep ocean. The actual mechanisms involved
in its formation and the organisms responsible is still currently unknown
although there is evidence for direct exudation by phytoplankton and also
production associated with the processing of DOM by heterotrophic bacteria.
Insight on the actual chemical compounds responsible is still lacking and
this contribution provides some valuable evidence in this light.

The contribution is broken into several parts. First the authors present
results from cultures with *Synechococcus*. The FDOM accumulated in the
cultures has similar fluorescence properties to marine DOM although this
comparison could be strengthened to be more convincing (see point 1). Next
they compare the dynamics of FDOM accumulation and development in
*Synechococcus* cell abundance. Here I am not convinced that there is a clear
direct link to *Synechococcus* (see point 2). How can the processing of
*Synechococcus* released DOM, by heterotrophic bacteria be ruled out here?
Next they provide additional optical data in the form of apparent quantum
yields and effects of photochemical exposure. These data, in my opinion,
although of good quality and interesting in their own right, do not support
the case they are trying to make. (See point 3). Finally the authors present
the result of the chemical characterization of the DOM accumulated in the
cultures and this leads them to conclude a significant presence of pyrrole
structures. These likely have their origins in plankton pigments and can be
expected to be a ubiquitous source in the ocean. They carry out
measurements on commercially available phycocyanobilin and find that
when photodegraded it has an optical signature similar to that of humic
marine DOM. This is a substantial finding and can easily be further
cemented by including the AQY spectra (See point 4).

In its current form I do not find the manuscript suitable for publication. Yes I
am convinced that phytoplankton pigments are the ultimate precursor
material for FDOM but is there a reason to expect that the FDOM precursor
material released by picocyanobacteria is specific to them and not released
by other organism (phytoplankton at least)? It is clear that some subsequent
extracellular abiotic and biotic processing of this material is required. The
evidence presented does not provide a convincing argument for why
picocyanobacteria should dominate the production, rather than this being a
more widespread source.

The argument that picocyanobacteria FDOM is highly similar to marine DOM has been
further strengthened by 1) adding a comparison with published datasets in openfluor and
the convincing match of our picocyanobacteria-derived PARAFAC components with
several previously published marine FDOM studies; 2) We added a figure 3 (and moved
the apparent quantum yield figure into the supplementary section) to show the single
wavelength excitation (250 nm) and emission (450 nm) spectra of the previously defined
“humic-like” component. This new figure underlines the close match of
picocyanobacteria-derived FDOM and deep-sea FDOM. We added references that
showed some contribution of FDOM from primary producers, but it was only estimated
to be max. 20% and not including the max. Fluorescence at Ex/Em 250/450 nm, our
FDOM signals suggest in comparison a much bigger contribution and much better
spectral match arising from picocyanobacteria. We do not want to claim that other
organisms do not contribute, we rather wanted to show a possible direct link of a FDOM
source (picocyanobacteria) and marine FDOM. We agree that the presented data do not
quantify FDOM possibly produced by picocyanobacteria and that was not our goal. This
study is aimed to convince the marine biogeochemical scientists that picocyanobacteria
possibly are important and significant contributors to marine FDOM. We rephrased some
of the sentences to better address that picocyanobacteria are widespread contributors to
marine FDOM but that we cannot assess at this stage if it is a dominant source of marine
FDOM.

In this work we found high similarity between FDOM signals from *Synechococcus*
culture and deep sea. Further investigations showed similar photodegradation behavior
and apparent fluorescent quantum yields as well. NMR pointed toward pyrrolic
substructures and FT-ICRMS was able to show dominant nitrogen molecules containing
two nitrogen atoms as potential FDOM molecules. The results indicated a strong
relationship between in situ deep ocean fluorescence and picocyanobacteria FDOM.
Picocyanobacteria are widespread in the world ocean and so is the “humic-like” FDOM
signal with no satisfactory explanation of in-situ sources; hence our study and we claim
that picocyanobacteria can be one of the important sources of autochthonous ocean
FDOM. However, we can hardly rule out the contribution of other phytoplankton
pigments based on the evidence we provided here, however, other studies had already
emphasized the contribution of picocyanobacteria to sinking particle flux. These could
provide circumstantial evidence for our assumption that picocyanobacteria contribute
significantly but likely not uniquely to the ocean FDOM signals, especially the
“humic-like” components accumulated at depth. Bacteria also play important roles in the

degradation of marine DOM. However, our incubation experiments with coastal seawater
showed that there was only a low level of “humic-like” fluorescent components in coastal
seawater without the addition of DOM from the *Synechococcus* culture. **Bacterial**
**abundance dynamic correlated with the production of “protein-like” components,**
**but not with the large Stoke’s shift of the “humic-like” fluorescent signal. The**
**bacteria activities triggered by *Synechococcus* FDOM caused likely “transformation”**
**and not “direct production” (see response to point 2) of FDOM.** Please see below
more specific responses to some very relevant questions raised.

Lines 110-116. It is difficult to follow this comparison with the way the
fluorescence data are currently plotted (contour plots). It would be easier if
the authors selected a "slice". This could be a specific emission spectrum or
even a diagonal which captures the main features (e.g. synchronas scan).
Space is at a premium, but the contour plots could be made smaller for
example as an inset to a 2d plot of the slice. Finally the comparison to
bacteria is also difficult as there is such a large peak at Em350. I wonder if
the scale is zoomed in maybe the humic-like signal with large Stokes Shifts
would appear.

We added another figure (Fig. 3) that nicely showed the excitation and emission spectra
at the humic-like fluorescence (Ex: 250 nm / Em: 450 nm). We normalized the intensity
to the maximum intensity of each signal at these specific excitation and emission
wavelengths. We moved the quantum yield figure to supplementary to accommodate this
figure. However, we feel that the contour plots are important to remain in Figure 2,
because most people working with marine FDOM are very familiar with these plots. The
heterotrophic bacterial FDOM showed only weak fluorescence at 450 nm and the
humic-like signal was not present in this sample. The new figure clearly showed that the
heterotrophic bacterium did not produce the specific “humic-like” component.

Line 121-133. This comparison is also difficult to follow. Again a 2D plot
like those in FigS2 and 3 would be better. I suggest replacing Fig 2 with a
2D plot of the excitation and emission loadings of each component
compared. The match would then be clearer to the reader. I think it would
also be apparent (as is from looking at the EEMs) that the emission spectra
match very well, but the marine DOM excitation spectra are broader.

Expand on your comparison to PARAFAC components from earlier studies
by including these components on plots (in sup info). These data should be
available on OpenFLuor.org

EEM-PARAFAC components are most often presented as contour plots and we would
like to keep this presentation, but the point raised is well taken. We added another figure
(Fig. S4) to show a comparison of our SYN1-3 PARAFAC components with published
datasets available in openfluor. Several previously published marine data sets matched
closely our three different components in both excitation and emission and underlined the
close spectral match of components between *Synechococcus* and marine FDOM.

Lines 134-141. It is stated that the increase in fluorescence in the non-virus
amended *Synechococcus* culture (Fig 3) was linked to a decrease in cell
abundance (FigS4). I can not see the decrease in cell abundance in the
culture without virus. In addition it is difficult to evaluate the changes in
bacteria numbers as the scale is too large. Bacteria numbers should be around
a factor of 100 lower. In fact the greatest increase in FDOM is seen between
135 day 4-7 and this is where it looks like there are greatest bacteria abundances.
The FDOM data for the virus amended treatment is not shown (as in Fig3,
only as a contour) but does this show the same pattern? Also it seems
strange when comparing the contour plots in Fig S4 with the Fig3. For
example day 7 has a low fluorescence (<0.25) but SYN1 in Fig3 is above 0.3.
I think something is mixed.

Fig.3 was replaced now with Fig. 4, showing the accumulation of fluorescent intensities
of PARAFAC modeling components in the *Synechococcus* culture. It can be now clearly
seen that the cell abundances of *Synechococcus* matched well the fluorescence and the
effect the virus had on the cell abundance and consequently the lysed cells.

Line 159-171. The comparison of AQY is a great idea but it is difficult to
follow the comparisons made. The AQY spectra in Fig4 appear very
different from each other.

The apparent fluorescence quantum yields (QY) are now in supplementary (Fig. S7). This
paragraph was rewritten to explain the data in more detail. The marine FDOM QY is not
very different at all from that of the picocyanobacteria and showed overall low values of
both. This fact alone is supporting that picocyanobacteria derived FDOM could be indeed
contributing to marine FDOM. If the values would be very different, then it could be
argued that different chromophores must be present. Even the local maxima of the
deep-ocean fluorescence are somewhat close to that of the cyanobacteria FDOM (only a
shift of about 10 nm). The major point here is that terrestrial-derived and
bacterial-derived FDOM is considerably lower and more different in QY than the
picocyanobacteria FDOM.

Line 175-178. The photodegradation results are very interesting but again it
is difficult to follow the authors comments when looking at Fig S6. The
clearest pattern apparent is that the FDOM loss is directly linked to the
original FDOM signal. So the SRNOM has longer wavelength fluorescence
and therefore also shows a loss of longer wavelength fluorescence

Direct photolysis is indeed directly related to the chromophores present, and hence
similar photochemical degradation would imply some similarities in molecular
composition of the initial chromophores. In analogy to the QY measurements, a very
different photochemical degradation between marine FDOM and picocyanobacteria
FDOM would have indicated different chromophores. A very similar photochemical
behavior must be present if any similarities are suggested, but similar photochemical

behavior is still not a proof for presence of the same chromophores. We added these
experiments to further strengthen the argument that picocyanobacteria FDOM is similar
to marine FDOM in its optical properties and its photochemical behavior. It is intriguing
that a presumably very complex marine FDOM and terrestrial DOM have such similar
photochemical degradation when compared to DOM derived from single organisms. This
suggests that marine FDOM can be potentially explained by a rather limited number of
chromophores.

Line 179-205. This is a very interesting result. Can the authors include the
AQY spectrum of phycocyanobilin before and after photodegradation. Is it
also possible to include the other chemical measurements (MS, NMR). This
would make for a very convincing argument.

We added the QY of phycocyanobilin before and after photodegradation and confirmed
the dramatic increase in QY induced by photolysis. Unfortunately, it is not possible to
add detailed FT-ICR-MS and NMR data to this set of photochemical experiments due to
the very low amount of DOM present, which restricts detailed molecular characterization.

Other points

Line 119. This sentence stands alone. What statistical analysis?

“Statistical analysis” here refers to the PARAFAC analyses, a clarification was added.

Line 142-147. I do not agree completely. The experiment has shown that the
combination of *Synechococcus*, heterotrophic bacteria and virus can result in
the production of FDOM signal, which is similar to marine DOM (in
emission in particular).

The viral-lytic process caused the FDOM release from *Synechococcus* cells (clearly
visible in Fig. 4). The bacteria-like signals were diagnosed in *Synechococcus* culture by
flow cytometry. Since it is difficult to maintain an axenic culture of *Synechococcus*
CB0101, we transferred fresh cultures of CB0101 frequently to keep the abundance of
heterotrophic bacteria low in cultures.

In this study, the abundance of heterotrophic bacteria remained at a low level, and did not
respond to the DOM released by *Synechococcus* cells. We added a discussion at lines
194-219.

Line 156-159. This statement is a little too bold. Dark yes but temperatures
were higher and the over all substrate levels were also very different. Coastal
water has higher background DOM levels.

Correction made.

The coastal water does have higher background DOM levels. However, in this
experiment, we used PARAFAC analyses of FDOM samples from the seawater
incubation with and without *Synechococcus* DOM addition. The three PARAFAC

components were diagnosed in the control group without *Synechococcus* DOM addition,
but at very low levels. The incubation results demonstrated the stability of *Synechococcus*
FDOM under dark and over bacterial degradation over 3 months.

Figure 1. It s unclear from the legend that the a & b are also SPE-DOM. Is
explained in the text (line 110-111).

Correction made.

Reviewer #2 (Remarks to the Author):

I find this short yet complete manuscript to be very interesting and novel. It
crosses some interesting methods (fluorescence EEMs, NMR and
FT-ICR-MS) to try to resolve the question of where deep sea FDOM comes
from. Here the authors propose that a proportion of the deep sea FDOM is of
picocyanobacterial origin (*Prochlorococcus* and *Synechococcus*). I find the
paper to be of an acceptable quality for Nature Communications and have
some small suggestions that will help to improve the clarity for the reader of
the article.

I find the abstract to be clear, well written, appropriate and easy to follow.

Page 3, line 105: when the authors say pure, do they mean axenic or that
they are non-axenic monocultures?

*The initial *Synechococcus* culture (strain CB0101) was non-axenic monoculture. To*
*address this issue we added an axenic culture and the corresponding EEM is now in the*
*supplementary information. No change of the humic-like component was observed,*
*however, the protein-like fluorescence disappeared, which has been often correlated to*
*heterotrophic bacteria in line with our results shown for the DSS3 bacteria culture.*

Line 106: where these measurements conducted on extracted or just filtered
samples?

*The EEMs in Fig.1 showed the SPE extracted samples for all the tested samples. The*
*samples used in the PARAFAC analysis from the 15 days viral-addition experiment of*
**Synechococcus* and 90 days seawater incubation experiment were undertaken on filtered*
*samples only. Correction made in the method of the new version (lines 423-428).*

Line 112: I agree that the spectra are similar but I think that the authors
should tone down the phrase 'striking congruence". Also, what about surface
water, does it look the same? Also, add in the letters for the deep ocean
SPE-DOM.

*Thanks for the suggestion, correction was made and we toned down on this sentence.*
*“Humic-like” fluorescence is drastically photobleached in the surface ocean and increases*
*with depth until steady levels are reached (see references within manuscript) and that is*
*the reason why we used deep ocean SPE DOM to compare it with picocyanobacteria.*

115: did the bacterial culture have algal DOM as a carbon source?

*No. This bacterial culture was cultured in YTSS medium (4 g l⁻¹ yeast extract, 2.5 g l⁻¹*
*tryptone, 20 g l⁻¹ Crystal Sea) at 28°C (see “Materials and Methods”)*

118: yes, but this does not prove that all deep ocean FDOM is on
picocyanobacteria origin.

Agreed. We assumed that picocyanobacteria to be one of the important sources of marine
autochthonous ocean FDOM production. Evidence in other studies showed that
picocyanobacteria contributed highly to sinking particle flux (see references in
manuscript), and our seawater incubation experiment also showed the stability of
*Synechococcus* FDOM under dark conditions. We explained it now in more detail in the
manuscript and that we are not trying to say that all deep-sea FDOM is originating from
picocyanobacteria.

167-169: yes, I agree that there are some nice parallels here between the
quantum yields from the picocyanobacteria FDOM and the deep ocean
sample. How do the authors explain the shifts? FDOM from other sources?
Aging of the FDOM? Do the authors have any thoughts on how the spectra
from other autotrophic sources might compare?

Apparent fluorescent quantum yield measurements of marine FDOM are rare and a
conclusive comparison with other FDOM samples cannot be made at this point. However,
this should be part of future investigations.

We tested FDOM in Arctic diatom cultures in the past, but no significant fluorescence
was found. However, previous studies had shown the variations of low level FDOM
signals in different diatom species. We added some references that showed low level
signals but no significant levels that represented the “humic-like” fluorescence and its
large Stoke’s shift. We added some discussion in the revised manuscript.

Page 15, line 14: did the authors acidify the samples for the fluorescence
spectroscopy analysis? If so, this will have had a negative effect on the
fluorescence measurements. See : Green and Blough (1994).

The samples were only acidified during the SPE procedure. The methanolic extracts were
then dried and re-dissolved in pure water (see method section), which resulted in a pH of
about 6.5. All samples that were compared in this study had roughly the same pH. We are
well aware of the pH effect on optical properties and published a paper on the effect of
pH on photochemistry (Timko et al., 2015).

Page 15, line 18. Why was air bubbling not used in all of the cultures?

Gentle air bubbling was used to maintain the mixing and provide sufficient air (mainly
CO₂) to a large volume (1 L) of *Synechococcus* culture. For the cultures in smaller
volume (50ml), there is sufficient air in the flasks. These cultures were shook to mix once
a day.

Line 29: coastal (check spelling)

Correction made.

304 Pg 16, line 52. Can the authors check the values for emission and excitation?
Perhaps I am missing something, but it appears that the emission values
were lower than the excitation values.

We checked this and it seems fine. We are not sure what was meant and did not find a
mistake here.

I am wondering if the extraction process altered the DOM and hence the
FDOM signals. Can the authors comment on this? Also as much work have
been published on non-extracted FDOM, I think it is important that the
authors clearly stated that their measurements were conducted on extracted
samples on the graphs (it is noted in some of them but not all, eg. Fig. 4).
This is a potential confusion point and it should be clearly dealt with in the
figure legends and in the text.

All samples in this study were solid phase extracted and details were added to the caption
of the figures. The FDOM is quantitatively recovered by the used SPE method and we
already demonstrated in a previous study that SPE-DOM reflects the same typical FDOM
trends throughout the water column in the Sargasso Sea (See Timko et al., 2015). It is
true that SPE somewhat fractionates DOM, but FDOM is quantitatively recovered. The
reason for this is that FDOM is considered to be relatively hydrophobic due to the
requirement of aromatic and conjugated π -electron systems. The used PPL resin is
perfectly suited to quantitatively capture these compounds.

Fig. 1. Please put the same y axis scale on the absorption graphs. Why are
there horizontal lines on the figures? This should be explained in the
legends.

The lines were not intentional. This probably happened during the conversion to pdf. We
scaled all absorbance spectra to the same scale and replaced the figures.

Fig. S4. For ease of comparison, it would be nice to have the same y axis
scaling on the cell abundance graphs.

Correction made in the revision Fig. 4.

Fig. S5. Please check the format of the upper figure, it looks a bit truncated.
Maybe change to "lower" rather than "bottom" when referring to the panels
in the graphs (here and elsewhere in the manuscript).

Correction made.

There is no use of statistics in this work. It would be good to have some idea
of the replicability of the measurements however. plus how many replicate
measurements were made for each analysis, etc.

All samples were run in triplicates, the error bars are given in appropriate figures. FT-MS
samples were also run in triplicates and showed similar results. The PARAFAC approach
is by its own right a statistical tool.

I find the conclusion to be robust and to be based on the results presented.

I think the format of the journal names needs to be checked, some are in full
whereas some are given as the abbreviated form. Please also check ref 1 of
the methods section (SYnechococcus)

Please check the format of Ref. 10 : the name should be corrected to:

Romera-Castillo, C.

Green, S.A., Blough, N.V., 1994. Optical absorption and fluorescence
properties of chromophoric dissolved organic matter in natural waters.

Limnol. Oceanogr. 39, 1903-1916.

**Correction made.**

Reviewer #3 (Remarks to the Author):

The authors present a clear and compelling argument for cyanobacterial
pigments contributing to the deep-ocean pool of fluorescent dissolved
organic matter. This is a significant result as this work clearly establishes a
pathway for an autochthonous source for CDOM components which has
only been inferred in the past. This in turn ties into questions of the origin of
deep-sea DOM and its lifetime, which are largely open to question. So I
think this topic is very important. The authors use state of the art techniques
in analysis and have combined observation and experiment in a logical way.

I would like to identify a couple areas in which the authors can strengthen
their argument. First, it seems to me there's a bit of a jump from
"tetrapyrrole" and "phycobilin" here. Phycobilins are not the only
tetrapyrroles in the marine environment, and in fact it can easily be argued
that they are far outnumbered by those found in chlorophylls and
cytochromes (for example). The authors must have convinced themselves
that the chemical structures they were seeing in the environment came from
phycobilins and not from other chemical structures containing tetrapyrroles
such as chlorophyll or cytochrome degradation products. Some clarification
here, I think, is essential.

We agree to the comment, but water-solubility plays a major role and most tetrapyrroles
in the environment are not readily water soluble. However, water soluble metabolites can
still be formed. Hence, we conducted experiments using commercially available
phycocyanobilin and found that photo-products of this chromophore resembled strong
similarities to the photochemical behavior of the DOM released by picocyanobacteria.
Water soluble chlorophyll metabolites such as Chlorin-P do not have that behavior (not
shown in this study).

As the reviewer said, there are other tetrapyrroles in the environment, we can hardly rule
out the sources of other origin but rather want to make a strong case that
picocyanobacteria are likely one source and possibly an important one. In the manuscript
we discussed the contribution of picocyanobacteria to the vertical POC flux, and the
detection of *Synechococcus* pigments in the deep ocean (see the references in the
manuscript). This fact indicated that *Synechococcus* cells were sinking to deeper depth.
We claim that it can be an important but not unique source of marine FDOM.

Also I think a bit more discussion is warranted on the differences (or lack
thereof) between the spectra of terrestrial humic material and the
cyanobacterial derived material. It is certain that there is some terrestrial
humic material in the deep ocean, and the question is how much compared
to the autochthonous fraction. The fluorescence signature of terrestrial

humic material is highly conserved. Is the similarity between these
substances of differing origin just a coincidence?

The already demonstrated depth profiles of FDOM in the world oceans highly suggest
that terrestrially-derived FDOM is not likely contributing significantly to the observed
FDOM distribution. A strong correlation of FDOM and AOU has been observed, but this
relationship seems to be weak in the Atlantic, which fueled the controversy about a direct
link between AOU and FDOM. We focused in this manuscript on the evidence we have
that marine FDOM is similar to picocyanobacteria FDOM, which is a conceivable source
material. The QY measurements suggested that marine FDOM is quite different to
terrestrially-derived DOM and that picocyanobacteria FDOM is much more similar to
marine FDOM than is terrestrially-derived FDOM. We believe that it is indeed a
coincidence which probably triggered the initial believe that deep-sea marine FDOM
largely originated from terrestrial-derived material, which seems not to be the case.

I have some minor quibbles regarding the use of GF/F filters to separate
dissolved fractions from particles (DON'T DO THIS), and with the
unquestioning use of SPE to concentrate dissolved organic matter. These
processes are selective, and the impacts of this selection should be
acknowledged.

GF/F filters were used because they can be baked to have no carbon contributing to the
DOM. We do however agree that some colloidal material and bacteria may pass through
the 0.7 μm filters. This was not a concern in this study, due to the fact that all samples
were run over the SPE resin and any remaining colloidal material would have been
trapped at the surface of the resin. We also did not observe colloidal material in our initial
samples, because no increased scattering was observed in the fluorescence when direct
measurements on just filtered water samples were undertaken. This can sometimes
happen when fine colloidal material is present (e.g. Fe precipitation etc.). FDOM is
quantitatively retained by the used resin, which makes SPE a very useful tool to
concentrate FDOM. We published a paper on SPE-DOM from the Sargasso Sea last year
showing that the trends observed between direct water samples and our SPE-DOM were
essentially the same (Timko et al, 2015). However, we were able to have a much
enhanced signal to noise ratio when SPE was used.

Reviewers' Comments:

Reviewer #2 (Remarks to the Author)

I find that the authors have adequately and clearer responded to my comments

Reviewer #3 (Remarks to the Author)

I am not entirely satisfied with the authors' responses to my earlier comments. I feel, upon consideration of their responses to the comments, that they have not devoted sufficient time to alternate hypotheses other than cyanobacterial pigments as the source of the new chromophores they have discovered in the humic category. There are plenty of ways solubility can be modified, and microbial processing of other tetrapyrrole forms can not be ruled out.

However, unless the editors disagree, I do not feel this precludes publication of the manuscript. In fact as scientists we advance most reliably when we propose hypotheses that are subsequently refuted. I feel this is a good hypothesis that deserves consideration, and as such I feel publication is warranted. As I said before this is a very clear, good study with wide implications in the field so I feel this is appropriate for Nature Communications.

Reviewer #4 (Remarks to the Author)

First of all, I would like to point out that the Editor approached me because reviewer #1 of this manuscript had to withdraw from this peer review process. The Editor was specifically interested on my assessment on how the authors have reacted to the concerns raised by that reviewer. In any case, I have carefully read the replies to the three reviewers and the revised version of the manuscript too.

As the three reviewers have already stated, the manuscript deals with an issue of high interest for the wide community of marine biologist and geochemist working on the ocean biological and microbial carbon pumps. The mechanisms of autochthonous production of fluorescent dissolved organic matter (FDOM) are still on debate and this manuscript adds clear evidences of a new source from *Synechococcus*.

Although the authors said that it was not their intention, *Synechococcus* is still presented as an "important and significant" source of DOM in the deep ocean. However, the authors only have evidences of the production of FDOM by *Synechococcus* but neither on its extend nor on its contribution to the current FDOM signal in the deep ocean. The terrestrial source (e.g. Andrew et al., *Marine Chemistry* 148, 33-43, 2013), the phytoplankton source (e.g. Romera-Castillo et al., *Limnology and Oceanography* 55, 446-454, 2010; Fukuzaki et al., doi:10.1093/plankt/fbu015, 2014) or the in situ microbial respiration source (e.g. Kramer and Herndl, *Aquatic Microbial Ecology* 36, 239-246, 2004; Yamashita and Tanoue, *Nature Geosciences* 1, 579-582, 2008; Jorgensen et al., *Marine Chemistry* 126, 139-148, 2011; Catalá et al., doi: 10.1038/ncomms6986, 2015) have been invoked as likely FDOM sources up to now. But, how much does this new *Synechococcus* source could represent? Following Reviewer #1, I think that the authors have to relax their enthusiasm concerning the relevance of this source to deep ocean FDOM. Furthermore, the authors should realise that if *Synechococcus* was not a quantitatively relevant FDOM source, maybe this article does not merit publication in a broad-audience but in a more specialised journal.

Concerning the four points raised by reviewer #1, I consider that, in general, the authors have taken into account her/his recommendations and modified the revised manuscript accordingly. However, I have some comments to add. Specifically, the way of presenting the emission spectra on Figure 2 of the revised manuscript (normalised to the maximum intensity) does not

demonstrate that the heterotrophic bacterium do not produce the specific humic-like component. The relative emission intensity of the bacterium at 450 nm is about 0.3 normalised units. Given that the maximum emission intensity of this culture is about 11 absolute units, this means that the intensity at 450 nm would be about 3 absolute units. This is much higher than the emission intensity of *Synechococcus*, *Prochlorococcus* and deep ocean DOM. Therefore, contrary to what you state, bacteria are indeed the major FDOM producers in your experiments!

Concerning point 2, although in the experiment conducted in this manuscript with coastal water, bacteria had produced low levels of FDOM, there is abundant literature concerning the production of humic like FDOM by marine bacteria (e.g. Kramer and Herndl, *Aquatic Microbial Ecology* 36, 239–246, 2004; Lonborg et al., *Marine Chemistry* 119, 121–129, 2010; Jorgensen et al., *Geophysical Research Letters*, doi: 10.1002/2014GL059428, 2014). Even the *Ruegeria pomeroyi* culture of the authors produced the highest amounts of humic like FDOM (see previous paragraph).

Concerning point 3, contrary to reviewer #1, I think that the experiment on photochemical exposure should be kept in the manuscript because they contribute to explain the link between FDOM originated by *Synechococcus* and deep water FDOM.

Finally, as reviewer #2 and #3, I am concerned above the comparability of the EEMs collected directly from seawater and those from the material retained by SPE on PPL resins. The authors rely on the recent paper by Timko et al. (*Frontiers in Marine Science* 2, doi:10.3389/fmars.2015.00066, 2015) to state that there was a quantitative recovery of FDOM with these resins. As far as I understood, Timko et al. (2015) just indicate that the same kind of EEMs were observed in seawater and the extracts, but quantitative retention is not demonstrated. Quantitative retention would mean obtaining the same fluorescence intensity in the original and extracted sample after concentration factor correction, and this seems not to be the case. So, please, relax again your statement concerning this point.

Responses to reviewer comments

Reviewer #3 (Remarks to the Author):

I am not entirely satisfied with the authors' responses to my earlier comments. I feel, upon consideration of their responses to the comments, that they have not devoted sufficient time to alternate hypotheses other than cyanobacterial pigments as the source of the new chromophores they have discovered in the humic category. There are plenty of ways solubility can be modified, and microbial processing of other tetrapyrrole forms cannot be ruled out. However, unless the editors disagree, I do not feel this precludes publication of the manuscript. In fact as scientists we advance most reliably when we propose hypotheses that are subsequently refuted. I feel this is a good hypothesis that deserves consideration, and as such I feel publication is warranted. As I said before this is a very clear, good study with wide implications in the field so I feel this is appropriate for Nature Communications.

Response to reviewer: We added an additional discussion on alternative sources and of course the reviewer is right to state that microbial processing of other sources of tetrapyrroles may also yield “humic-like” FDOM. But thus far no evidence is available that metabolites of chlorophyll or other pigments will yield similar fluorescence pattern and we did not find any evidence in the literature pointing towards other widespread marine sources of this rather specific FDOM. We are also only suggesting that it could be pyrrolic, but the chromophore itself still remains unknown and warrants further research, which hopefully will be triggered by this publication. However, we hope we extended the discussion in a meaningful and informative way to further discuss alternative pyrrolic sources, but the main focus was that a specific source of “humic-like’ FDOM was found, that has the potential to explain at least in part marine FDOM.

Reviewer #4 (Remarks to the Author):

First of all, I would like to point out that the Editor approached me because reviewer #1 of this manuscript had to withdraw from this peer review process. The Editor was specifically interested on my assessment on how the authors have reacted to the concerns raised by that reviewer. In any case, I have carefully read the replies to the three reviewers and the revised version of the manuscript too.

As the three reviewers have already stated, the manuscript deals with an issue of high interest for the wide community of marine biologist and geochemist working on the ocean biological and microbial carbon pumps. The mechanisms of autochthonous production of fluorescent dissolved organic matter (FDOM) are still on debate and this manuscript adds clear evidences of a new source from *Synechococcus*.

Although the authors said that it was not their intention, *Synechococcus* is still presented as an “important and significant “source of DOM in the deep ocean. However, the authors only have evidences of the production of FDOM by *Synechococcus* but neither on its extend nor on its contribution to the current FDOM signal in the deep ocean. The terrestrial source (e.g. Andrew et al., *Marine Chemistry* 148, 33-43, 2013), the phytoplankton source (e.g. Romera-Castillo et al., *Limnology and Oceanography* 55, 446-454, 2010; Fukuzaki et al., doi:10.1093/plankt/fbu015, 2014) or the in situ microbial respiration source (e.g. Kramer and Herndl, *Aquatic Microbial Ecology* 36, 239–246, 2004; Yamashita and Tanoue, *Nature Geosciences* 1, 579-582, 2008; Jorgensen et al., *Marine Chemistry* 126, 139-148, 2011; Catalá et al., doi: 10.1038/ncomms6986, 2015) have been invoked as likely FDOM sources up to now. But, how much does this new *Synechococcus* source could represent?

Response: To further emphasis the possible significance of FDOM released by lysed picocyanobacteria, we calculated the amount of FDOM (based on quinine sulfate equivalent mass) that can be supplied from *Synechococcus*. This calculation was based on field observations and flux calculations previously undertaken and on global annual abundance data. The following text was added to the manuscript:

*“Picocyanobacterial pigments (e.g. phycoerythrin, PE) also have been shown to be more readily exported down to deep ocean layers than chlorophyll or other phytoplankton pigments, according to sediment trap records⁹. In our laboratory culture experiment, the Fmax production rate of humic-like EEM-PARAFAC component (SYN1, Fig.4) was $3.71 \pm 1.84 \times 10^{-12}$ QSUM cell⁻¹. It should be noted here that the fluorescence was converted into equivalent mass of quinine sulfate (QSUM, see method section). According to sediment trap records from the Costa Rica upwelling dome, the total *Synechococcus* export to depth ranged between 0.04% and 1.06% of the standing stock of *Synechococcus*⁹. The annual mean global abundance of *Synechococcus* was also previously estimated based on greater than 35,000 observations to be $7.0 \pm 0.3 \times 10^{26}$ cells, respectively⁵². By using the percentage range (0.04%-1.06%) of exported *Synechococcus* cells to depth mentioned above, we can calculate a global annual humic-like fluorescence export ranging between 1.04×10^{12} and 2.75×10^{13} QSUM. This estimate is grossly underestimating the whole contribution from all picocyanobacteria to the biological pump, because *Prochlorococcus* was not included in this calculation, but underlines that *Synechococcus* alone has the potential to substantially contribute FDOM to the deep ocean.”*

Following Reviewer #1,

I think that the authors have to relax their enthusiasm concerning the relevance of this source to deep ocean FDOM. Furthermore, the authors should realise that if *Synechococcus* was not a quantitatively relevant FDOM source, maybe this article does not merit publication in a broad-audience but in a more specialised journal.

Response: We truly hope that the added estimate of FDOM that can be supplied to the deep ocean convinced the reviewer that it indeed has the potential to be quantitatively relevant source of deep-sea FDOM.

Concerning the four points raised by reviewer #1, I consider that, in general, the authors have taken into account her/his recommendations and modified the revised manuscript accordingly. However, I have some comments to add. Specifically, the way of presenting the emission spectra on Figure 2 of the revised manuscript (normalised to the maximum intensity) does not demonstrate that the heterotrophic bacterium do not produce the specific humic-like component. The relative emission intensity of the bacterium at 450 nm is about 0.3 normalised units. Given that the maximum emission intensity of this culture is about 11 absolute units, this means that the intensity at 450 nm would be about 3 absolute units. This is much higher than the emission intensity of *Synechococcus*, *Prochlorococcus* and deep ocean DOM. Therefore, contrary to what you state, bacteria are indeed the major FDOM producers in your experiments!

Response: This figure was intended to show the location of the local fluorescent maxima as requested by a previous reviewer and should not be taken as a quantitative comparison and the tailing of the protein-like peak will not produce a humic-like fluorescence peak as described in detail in the EEM data. However, we noticed a mistake in our data and corrected the fluorescence spectrum accordingly. The QSU unit was wrong by an order of magnitude when compared to the UV-Vis data and we are grateful that this was pointed out by the reviewer. Please see corrected QSU scale in Fig. 1.

Concerning point 2, although in the experiment conducted in this manuscript with coastal water, bacteria had produced low levels of FDOM, there is abundant literature concerning the production of humic like FDOM by marine bacteria (e.g. Kramer and Herndl, *Aquatic Microbial Ecology* 36, 239–246, 2004; Lonborg et al., *Marine Chemistry* 119, 121–129, 2010; Jorgensen et al., *Geophysical Research Letters*, doi: 10.1002/2014GL059428, 2014). Even the *Ruegeria pomeroyi* culture of the authors produced the highest amounts of humic like FDOM (see previous paragraph).

Response: We are not disputing that other sources of humic-like fluorescence may exist but no direct linkage to individual heterotrophic bacteria strains or even abundance have been given thus far and the produced FDOM mentioned in the literature appear to be rather small. For example, Kramer and Herndl 2004 described bacterial derived FDOM, but only a generic fluorescence in QSU is given. Joergensen et al 2014 described nicely a small increase in ‘humic-like’ fluorescence in 13 months of incubation, but could not constrain if

this humic-like FDOM was produced directly or indirectly by prokaryotes and again no direct link to bacteria abundance was made. Results presented by Lonborg et al 2010 also indicated an increase in humic-like fluorescence although at longer excitation wavelengths during 70 days incubation, but again no direct link to specific bacteria were made. However we included these studies now to emphasize that heterotrophic bacterial communities may also play a role but that the direct link of specific bacteria to CDOM production is not clear. We also added another chart to Fig. S6 that shows the EEM-PARAFAC components for the seawater blank and it can clearly be seen that they are much lower and also rather stable over time, except the protein-like fluorescence, which slightly increased at the beginning of the incubation.

Concerning point 3, contrary to reviewer #1, I think that the experiment on photochemical exposure should be kept in the manuscript because they contribute to explain the link between FDOM originated by *Synechococcus* and deep water FDOM.

Response: Unfortunately, we already have 5 figures in the main text and we would need to keep the photochemical results in the supplementary material.

Finally, as reviewer #2 and #3, I am concerned above the comparability of the EEMs collected directly from seawater and those from the material retained by SPE on PPL resins. The authors rely on the recent paper by Timko et al. (Frontiers in Marine Science 2, doi:10.3389/fmars.2015.00066, 2015) to state that there was a quantitative recovery of FDOM with these resins. As far as I understood, Timko et al. (2015) just indicate that the same kind of EEMs were observed in seawater and the extracts, but quantitative retention is not demonstrated. Quantitative retention would mean obtaining the same fluorescence intensity in the original and extracted sample after concentration factor correction, and this seems not to be the case. So, please, relax again your statement concerning this point.

Response: Our results did not show any differences between SPE-DOM and direct water measurements of filtered cultures. Our data is also in very good agreement with the global data set published by Jorgensen et al 2011. Furthermore, FDOM chromophores have to be aromatic and conjugated π -electron systems and hence fall within the optimal conditions to be adsorbed to the PPL resin. However, we did relax the statement, because extraction efficiencies were not directly measured on these samples.

Reviewers' Comments:

Reviewer #4 (Remarks to the Author)

I have read the reply of the authors to my comments and the associated changes in the manuscript and I feel satisfied with the job they did.